# The dual role of chloride in synaptic vesicle glutamate transport

Roger Chang[1,2,3], Jacob Eriksen[1,2], Robert H Edwards[1,2,3,4,5]*

[1]Department of Physiology, UCSF School of Medicine, San Francisco, United States; [2]Department of Neurology, UCSF School of Medicine, San Francisco, United States; [3]Graduate Program in Biomedical Sciences, UCSF School of Medicine, San Francisco, United States; [4]Kavli Institute for Fundamental Neuroscience, UCSF School of Medicine, San Francisco, United States; [5]Weill Institute for Neurosciences, UCSF School of Medicine, San Francisco, United States

**Abstract** The transport of glutamate into synaptic vesicles exhibits an unusual form of regulation by Cl⁻ as well as an associated Cl⁻ conductance. To distinguish direct effects of Cl⁻ on the transporter from indirect effects via the driving force $\Delta\psi$, we used whole endosome recording and report the first currents due to glutamate flux by the vesicular glutamate transporters (VGLUTs). Chloride allosterically activates the VGLUTs from both sides of the membrane, and we find that neutralization of an arginine in transmembrane domain four suffices for the lumenal activation. The dose dependence suggests that Cl⁻ permeates through a channel and glutamate through a transporter. Competition between the anions nonetheless indicates that they use a similar permeation pathway. By controlling both ionic gradients and $\Delta\psi$, endosome recording isolates different steps in the process of synaptic vesicle filling, suggesting distinct roles for Cl⁻ in both allosteric activation and permeation.

DOI: https://doi.org/10.7554/eLife.34896.001

*For correspondence:
Robert.Edwards@ucsf.edu

Competing interests: The authors declare that no competing interests exist.

## Introduction

The regulated release of classical neurotransmitters by exocytosis requires their transport into synaptic vesicles. A H⁺ electrochemical gradient ($\Delta\mu_{H+}$) generated by the vacuolar-type H⁺-ATPase provides the driving force for vesicular uptake of all classical neurotransmitters, and uptake of the principal excitatory transmitter glutamate depends primarily on the electrical component $\Delta\psi$ (*Naito and Ueda, 1985*; *Maycox et al., 1988*; *Tabb et al., 1992*). Although Cl⁻ entry dissipates $\Delta\psi$, low concentrations of cytosolic Cl⁻ nonetheless stimulate vesicular glutamate transport (*Naito and Ueda, 1985*; *Maycox et al., 1988*; *Tabb et al., 1992*; *Hartinger and Jahn, 1993*; *Wolosker et al., 1996*; *Juge et al., 2010*). Lumenal Cl⁻ has also been shown to stimulate glutamate uptake (*Schenck et al., 2009*; *Preobraschenski et al., 2014*; *Eriksen et al., 2016*), but it has remained unclear whether this reflects a direct interaction with the transporter or an indirect effect through regulation of the driving force $\Delta\psi$ (*Martineau et al., 2017*). Further, the vesicular glutamate transporters (VGLUTs) themselves exhibit a Cl⁻ conductance (*Bellocchio et al., 2000*; *Schenck et al., 2009*; *Preobraschenski et al., 2014*; *Eriksen et al., 2016*) that has the potential to increase the driving force $\Delta\psi$ when lumenal Cl⁻ is high, or dissipate $\Delta\psi$ when lumenal Cl⁻ is low (*Eriksen et al., 2016*; *Martineau et al., 2017*). Interestingly, the plasma membrane excitatory amino acid transporters (EAATs) that clear glutamate from the synaptic cleft also exhibit an associated Cl⁻ conductance. Glutamate activates the conductance associated with the EAATs, but these currents are not stoichiometrically coupled to transport, and Cl⁻ is not required for glutamate transport (*Wadiche et al., 1995*; *Arriza et al., 1997*; *Machtens et al., 2015*). The Cl⁻ conductance associated with the VGLUTs

suggests functional resemblance to the EAATs even though these two protein families show no similarity at the level of primary sequence or tertiary structure.

The available radiotracer and fluorescence-based flux assays have greatly limited our understanding of vesicular glutamate transport. Synaptic vesicles from the brain accumulate radiolabeled glutamate in vitro, but reliance on the $H^+$ pump, the presence of other proteins and shifting ionic gradients make it difficult to determine the properties of the VGLUTs from this native preparation. Heterologous expression of the VGLUTs does not confer the robust activity observed with native synaptic vesicles, forcing reliance on the reconstitution of purified protein into artificial membranes (*Juge et al., 2006*; *Schenck et al., 2009*; *Preobraschenski et al., 2014*). However, dependence of virtually all the available assays on a $H^+$ pump inextricably links the driving force $\Delta\psi$ to ionic gradients, making it impossible to disentangle the effects of $Cl^-$ on $\Delta\psi$ from direct effects on the transporter.

The electrophysiological analysis of transport-associated currents provides a way to circumvent these limitations. The independent control of $\Delta\psi$ in voltage clamp recordings provides a way to distinguish between ionic coupling and allosteric control by ions such as $Cl^-$. In addition, dependence on $\Delta\psi$ as the driving force indicates that the VGLUTs are electrogenic and hence capable of producing detectable currents. However, slow turnover may limit the magnitude of transport-associated currents, and the VGLUTs are almost entirely intracellular, limiting access to electrophysiological recording. In previous work, we used misexpression of internalization-defective VGLUTs at the plasma membrane of *Xenopus* oocytes to demonstrate allosteric activation of the VGLUT-associated $Cl^-$ conductance by external (lumenal) $H^+$ and $Cl^-$ independent of any effects of $Cl^-$ on the driving force that may occur under physiological conditions (*Eriksen et al., 2016*). However, we did not detect glutamate currents in this preparation, raising questions about the role of these mechanisms in vesicular glutamate transport. Since expression at the plasma membrane does not enable direct addition of glutamate to the cytosolic face of the VGLUTs, we have now used patch clamp recording from more physiologically relevant endosomal membranes.

## Results

### Whole endosome recording detects glutamate currents by the VGLUTs

Similar to other synaptic vesicle proteins, the wild type VGLUTs target to endosomes when expressed in non-neural cells (*Tan et al., 1998*; *Foss et al., 2013*). To produce enlarged endosomes amenable to the patch clamp technique, we cotransfected HEK293 cells with a constitutively active version of early endosome Rab5 (Q79L) as well as VGLUT fused to EGFP (*Stenmark et al., 1994*; *Dong et al., 2008*; *Cang et al., 2013*). After mechanical isolation of the enlarged $GFP^+$ endosomes, we performed whole endosome recording with a pipette solution containing 140 mM $Cl^-$ at pH 5 since this maximally activated the VGLUT-associated $Cl^-$ conductance in oocytes (*Eriksen et al., 2016*) (*Figure 1A*) and mimics the conditions in recycling synaptic vesicles. To minimize background currents due to endogenous conductances, we eliminated permeant cations from all solutions. We also included the $Cl^-$ channel blocker 5-nitro-2-(3-phenylpropylamino)benzoic acid (NPPB) in the pipette to inhibit endogenous $Cl^-$ currents since we had previously found that NPPB does not affect the VGLUT-associated $Cl^-$ conductance in oocytes (*Eriksen et al., 2016*). Using voltage ramps from −100 to +100 mV in endosomes expressing VGLUT1, we observe small but substantial inwardly rectifying currents in the absence of external $Cl^-$ (*Figure 1A,C*). In the absence of permeant cations, these currents most likely reflect the efflux of $Cl^-$. Controls expressing Q79L Rab5 alone show no inward currents due to the suppression of an endogenous $Cl^-$ conductance by lumenal (pipette) NPPB (*Figure 1B*, *Figure 1—figure supplement 1B,C*). The addition of external $Cl^-$ (140 mM) does not consistently affect the inward current but substantially increases the outward currents associated with VGLUT1, with no effect observed in Q79L Rab5 controls (*Figure 1A–C*). The small size of the currents enables the low background to interfere with the precise determination of reversal potential, but shifts in the appropriate direction are consistent with the behavior of $Cl^-$ as a permeant ion. The currents also vary in magnitude, which might reflect differences in the expression on individual endosomes (*Figure 1* insets). Analysis of the other two isoforms (VGLUT2 and 3) shows behavior similar to VGLUT1 (*Figure 1—figure supplement 2A–D*). It is possible that the isoforms differ in current amplitude but the selection of $GFP^+$ endosomes for recording makes it difficult to assess a role for

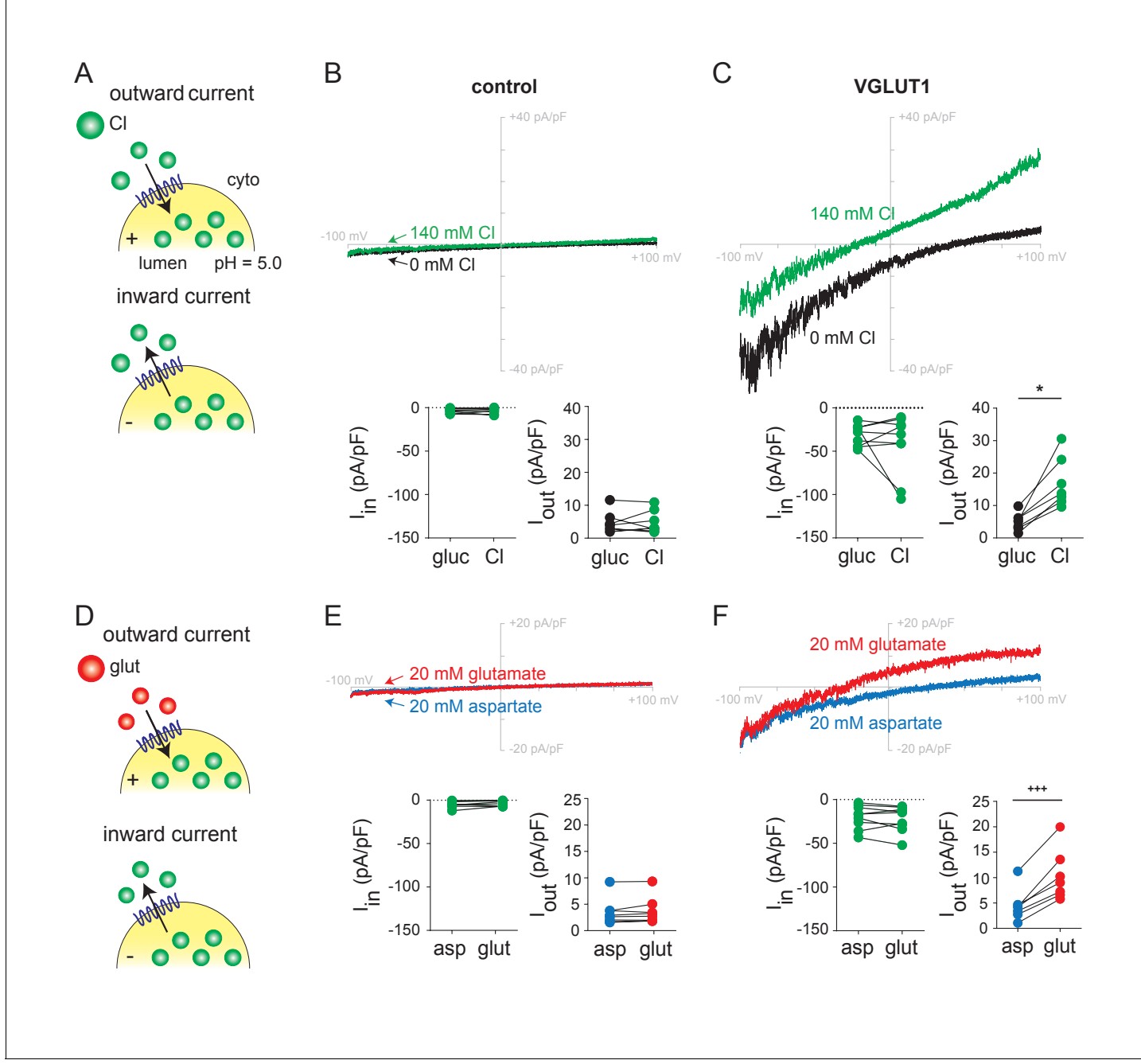

**Figure 1.** Chloride and glutamate currents in endosomes expressing VGLUT1. (A,D) Whole endosome patch clamp recording configuration, with 140 mM Cl⁻ at pH 5.0 and NPPB in the pipette, showing outward currents at positive lumenal potentials and inward currents at negative potentials. Representative recordings from control (Rab5 Q79L) and VGLUT1⁺ (Rab5 Q79L) endosomes with voltage ramps from −100 mV to +100 mV are shown from the same endosome in (B) as in (E) and in (C) as in (F), in the presence of external gluconate (0 Cl⁻), Cl⁻ (140 mM), aspartate or glutamate (both 20 mM). Insets show the maximum inward and outward currents for each endosome (B, $I_{in}$, p=0.698, $I_{out}$, p=0.748 both by paired t-test; C, $I_{in}$, p=0.910, $I_{out}$, p=0.0156 both by Wilcoxon; E, $I_{in}$, p=0.202, $I_{out}$, p=0.232 both by paired t-test; F, $I_{in}$, p=0.360 by paired t-test, $I_{out}$, p=0.001 by paired t-test, n = 7–9 for all). *p<0.05 by Wilcoxon matched pairs test; +++p<0.001 by paired t-test.

DOI: https://doi.org/10.7554/eLife.34896.002

The following figure supplements are available for figure 1:

**Figure supplement 1.** Chloride currents in endosomes without lumenal NPPB.

DOI: https://doi.org/10.7554/eLife.34896.003

**Figure supplement 2.** Chloride and glutamate currents (with lumenal NPPB) in VGLUT2⁺ and VGLUT3⁺ endosomes.

*Figure 1 continued*

DOI: https://doi.org/10.7554/eLife.34896.004

**Figure supplement 3.** Sensitivity of VGLUT1 Cl⁻ and glutamate currents to Evans Blue.

DOI: https://doi.org/10.7554/eLife.34896.005

differences in transporter expression. Like the VGLUTs expressed in oocytes (*Eriksen et al., 2016*), VGLUTs expressed in endosomes thus exhibit a Cl⁻ conductance.

To assess the flux of glutamate, we added the amino acid to endosomes expressing VGLUT1. Glutamate (20 mM) produces similar but slightly smaller outward currents than Cl⁻ added to the same endosomes, with a negative shift in reversal potential supporting behavior as a permeant ion (*Figure 1D–F*). To test specificity, we used the closely related amino acid aspartate, which is not recognized by the VGLUTs (*Maycox et al., 1988*; *Carlson et al., 1989*). In contrast to glutamate, aspartate produces no outward currents and cells expressing mutant Rab5 alone show no response to glutamate or aspartate (*Figure 1D–F*). VGLUT2 and 3 exhibit outward currents similar to VGLUT1 in response to glutamate and not aspartate (*Figure 1—figure supplement 2D–F*). Further, the dye Evans Blue, which is known to block vesicular glutamate transport (*Chaudhry et al., 2008*), inhibits both Cl⁻ and glutamate currents with submicromolar potency (*Figure 1—figure supplement 3*). The residual currents despite Evans Blue presumably reflect endogenous HEK cell conductances. Despite the difficulty inherent in recording transport-associated currents from intracellular membranes, we show that endosome recording can thus detect glutamate transport currents mediated by the VGLUTs.

## Allosteric activation of the VGLUTs by lumenal and cytosolic Cl⁻

To understand how Cl⁻ stimulates vesicular glutamate transport, we first used endosome recording to characterize the role of lumenal Cl⁻. As anticipated, reduction of the lumenal (pipette) Cl⁻ to 10 mM effectively eliminates the inward currents (due to Cl⁻ efflux) mediated by VGLUT1 (*Figure 2A*). However, the reduction also prevents outward currents due to both external Cl⁻ and glutamate, and 30 mM lumenal Cl⁻ has a similar effect (*Figure 2A,B*). At 50 mM lumenal Cl⁻, however, the outward currents due to external Cl⁻ and glutamate become more substantial, as do the inward currents due to Cl⁻ efflux (*Figure 2C*). Independent of effects on pH and Δψ, lumenal Cl⁻ is thus required to activate the flux of glutamate as well as Cl⁻, supporting the allosteric role proposed previously on the basis of the VGLUT-associated Cl⁻ conductance (*Eriksen et al., 2016*). The inward currents increase with lumenal Cl⁻, consistent with its role as permeant ion, but the outward currents saturate at 50 mM lumenal Cl⁻, consistent with the allosteric activation by lumenal Cl⁻ of the VGLUT-associated Cl⁻ conductance previously reported (*Figure 2—figure supplement 1*) (*Eriksen et al., 2016*).

We then examined the role of cytosolic Cl⁻, maintaining pipette Cl⁻ at 140 mM in light of the requirement for lumenal activation. In the absence of external Cl⁻, glutamate produces easily detectable outward currents (*Figure 1F*, *Figure 1—figure supplement 2D–F*). Consistent with this, synaptic vesicles do not absolutely require external (cytosolic) Cl⁻ to accumulate glutamate (*Disbrow et al., 1982*; *Naito and Ueda, 1985*; *Carlson et al., 1989*; *Hartinger and Jahn, 1993*; *Wolosker et al., 1996*). To determine whether external Cl⁻ has any impact on glutamate currents, we compared the effect of external Cl⁻ in the presence and absence of external glutamate (*Figure 2D*). At 10 mM, external Cl⁻ produces small outward currents just slightly larger than those with no external Cl⁻ (*Figure 2E*). In contrast, 10 mM Cl⁻ clearly potentiates the currents due to 20 mM glutamate (*Figure 2F,G*), and this larger effect in the presence of glutamate excludes a simple additive interaction between the two currents. With whole endosome voltage clamp eliminating any change in ionic gradients or Δψ, this potentiation strongly supports an allosteric mechanism for the acceleration of synaptic vesicle filling and glutamate release by cytosolic Cl⁻ (*Hori and Takahashi, 2012*).

We also used endosome recording to assess the recognition and permeation of other anions. Previous work has identified only one anion other than Cl⁻ that activates vesicular glutamate transport—Br⁻ (*Naito and Ueda, 1985*; *Eriksen et al., 2016*). By comparison to endosomes with low lumenal Cl⁻ (*Figure 2A*), we now find that lumenal Br⁻ activates glutamate currents in endosomes expressing VGLUT1-3 (*Figure 2—figure supplement 2A–D*). In addition, external Br⁻ produces outward

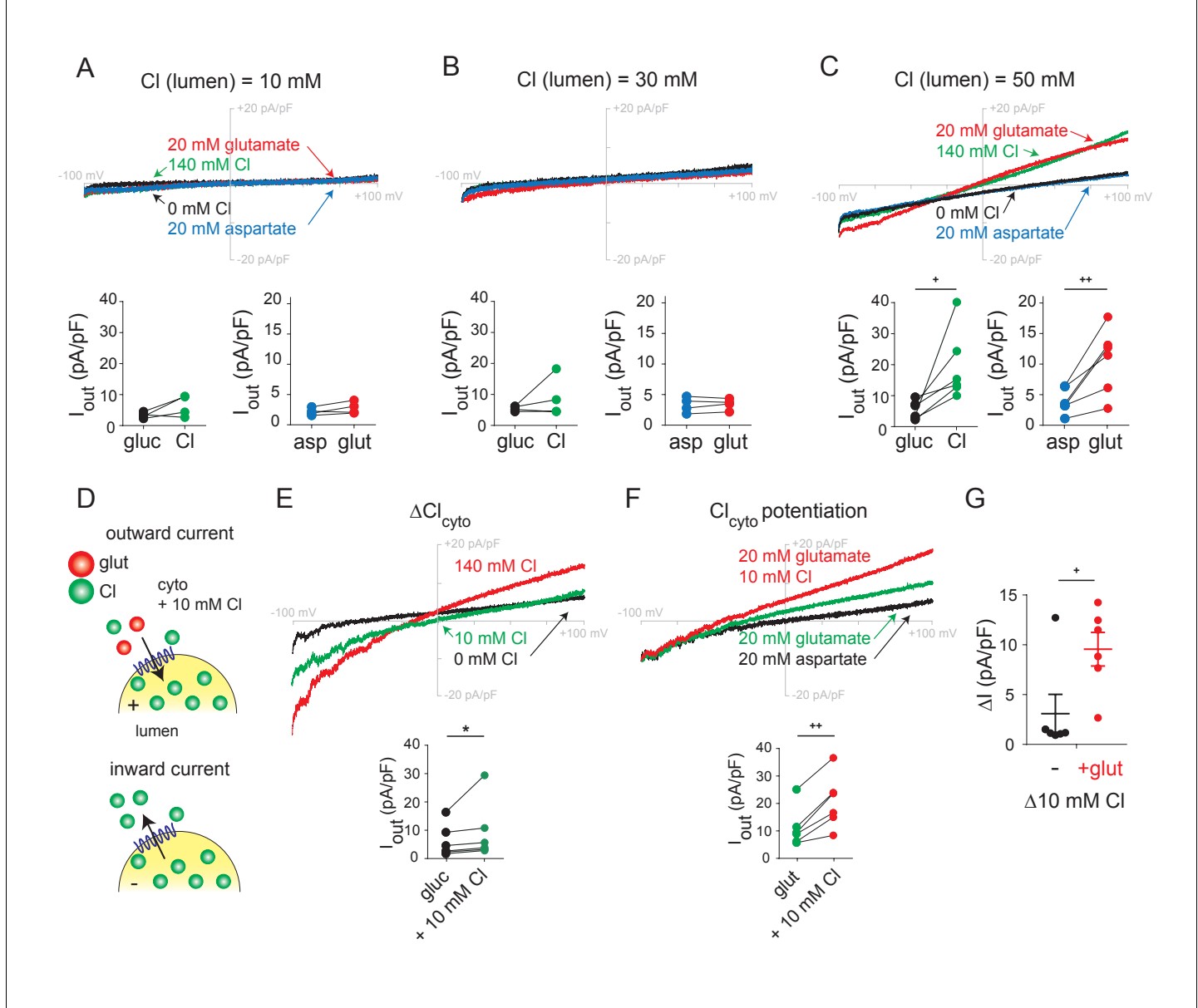

**Figure 2.** Role of lumenal and cytoplasmic Cl⁻ in endosomes expressing VGLUT1. (**A–C**) Representative whole endosome recordings under the same conditions as in *Figure 1* but in the presence of 10 mM (**A**), 30 mM (**B**) and 50 mM lumenal Cl⁻ (**C**) (n = 4 each). (**D**) To assess potentiation, 10 mM Cl⁻ was added to the external solution without (**E**) or with (**F**) 20 mM glutamate (n = 5 each). Insets show maximum outward currents for each endosome in the different external conditions (A, $I_{Cl}$, p=0.147, $I_{glut}$, p=0.150 both by paired t-test, B, $I_{Cl}$, p=0.327, $I_{glut}$, p=0.666 both by paired t-test, C, $I_{Cl}$, p=0.035, $I_{glut}$, p=0.010 both by paired t-test, E, $I_{Cl}$, p=0.01 by Wilcoxon, F, $I_{glut}$, p=0.002 by paired t-test). (**G**) Increase in outward current produced by 10 mM external Cl⁻ without or with 20 mM glutamate (p=0.034 by paired t-test). *p<0.05 by Wilcoxon; +p<0.05 and ++p<0.01 by paired t-test.

DOI: https://doi.org/10.7554/eLife.34896.006

The following figure supplements are available for figure 2:

**Figure supplement 1.** Dependence of VGLUT1 currents on lumenal Cl⁻.

DOI: https://doi.org/10.7554/eLife.34896.007

**Figure supplement 2.** Effects of Br⁻ and $PO_4^{3-}$ on VGLUT1⁺ endosomes.

DOI: https://doi.org/10.7554/eLife.34896.008

currents with a negative shift in reversal potential similar to that caused by Cl⁻ and glutamate, indicating that, like the other anions, Br⁻ also permeates. The VGLUTs were originally identified as Na⁺-dependent phosphate transporters (*Ni et al., 1994*; *Aihara et al., 2000*), and experiments with purified recombinant protein reconstituted into artificial membranes have supported this function (*Juge et al., 2006*; *Preobraschenski et al., 2018*). However, the addition of phosphate to endosomes expressing VGLUT1 did not produce any detectable currents in the presence or absence of external Na⁺ (*Figure 2—figure supplement 2E–G*). The VGLUTs may therefore either fail to transport phosphate, or mediate electroneutral cotransport of Na⁺ and phosphate).

## A conserved arginine residue in TM4 suffices for allosteric activation of the VGLUTs by chloride

To understand the relationship between allosteric activation and permeation by Cl⁻, we sought to identify mutations that distinguish between these functions, focusing on conserved basic residues in transmembrane domains (TMs) that might interact with Cl⁻ (*Figure 3A*). Taking advantage of structural modeling and previous functional studies (*Juge et al., 2006*; *Almqvist et al., 2007*; *Herman et al., 2014*; *Eriksen et al., 2016*), we first neutralized a highly conserved arginine residue in TM7. Consistent with previous work in *Xenopus* oocytes (*Eriksen et al., 2016*), all 3 VGLUT isoforms with this mutation (R314A in VGLUT1, R322A in VGLUT2 and R326A in VGLUT3) show no significant inward or outward Cl⁻ or glutamate currents (*Figure 3B,C*, *Figure 3—figure supplement 1A–C,D,G*), similar to control endosomes expressing Rab5 Q79L alone (*Figure 1B*). Importantly, this mutation does not impair VGLUT expression (*Figure 3—figure supplement 2F*) (*Eriksen et al., 2016*) or localization of the GFP-tagged protein (data not shown), indicating a specific requirement for this residue in transport.

We then focused on a conserved arginine in TM1. Replacement by alanine (R80A in VGLUT1) also eliminates the outward currents produced by Cl⁻ and glutamate entry (*Figure 3D*). However, this mutation spares the inward currents, indicating that the residue has a specific role in uptake of both anions but not Cl⁻ efflux. It is more difficult to evaluate the baseline inward currents due to Cl⁻ efflux than the outward currents due to anion addition, but all three isoforms (including R88A in VGLUT2 and R93A in VGLUT3) exhibit the same phenomenon (*Figure 3—figure supplement 1A–C,E,H*). Efflux of Cl⁻ may thus involve a mechanism distinct from glutamate translocation or even Cl⁻ entry. Alternatively, this arginine may affect rectification of the currents.

Third, we neutralized the conserved arginine in TM4 (Arg176 in VGLUT1, Arg184 in VGLUT2 and Arg189 in VGLUT3). As previously shown (*Figure 2*), both inward and outward currents require lumenal Cl⁻ (*Figure 3F*). Despite the absence of lumenal Cl⁻, R176A VGLUT1 allows robust outward currents due to the external addition of Cl⁻ or glutamate (*Figure 3G*), and R184A and R189A have the same effects in VGLUT2 and 3, respectively (*Figure 3—figure supplement 1A–C,F,I*). To test the role of the positive charge in recognition of Cl⁻, we replaced Arg176 in VGLUT1 by the similarly cationic amino acid lysine (R176K) or by neutral glutamine (R176Q). Like wild type VGLUT1, R176K exhibits currents in the presence but not the absence of lumenal Cl⁻ (*Figure 3—figure supplement 2*). However, the R176Q mutant does not exhibit currents even with lumenal Cl⁻, making it difficult to assess the impact on Cl⁻ recognition (*Figure 3—figure supplement 2C*). Western analysis indicated that all of the VGLUT1 mutants are expressed (*Figure 3—figure supplement 2F*); although the levels vary, we selected GFP⁺ endosomes for all of these experiments and the functional R176K mutant was expressed at levels similar to nonfunctional R80A and R176Q. To test further the role of Arg176 in Cl⁻ recognition, we took advantage of an internalization-defective form of VGLUT2 previously used to assess Cl⁻ currents across the plasma membrane in HEK cells as well as *Xenopus* oocytes (*Eriksen et al., 2016*). As previously shown, whole cell recording of HEK cells expressing internalization-defective but otherwise wild type VGLUT2 exhibit inwardly rectifying Cl⁻ currents that depend on high external Cl⁻ as well as low pH (*Figure 3—figure supplement 3*). In contrast, R184A VGLUT2 exhibits inward currents independent of external Cl⁻. In both whole endosome and whole cell configurations, dependence on Cl⁻ thus requires a cationic residue at this position in TM4, strongly suggesting that lumenal Cl⁻ normally binds at this site, and neutralization of this residue suffices to fulfill the allosteric activation required for both Cl⁻ conductance and glutamate transport. Anion permeation persists despite this mutation, further indicating that this residue is required only for allosteric activation by Cl⁻. Allosteric activation and permeation thus involve distinct interactions of Cl⁻ with the transporter.

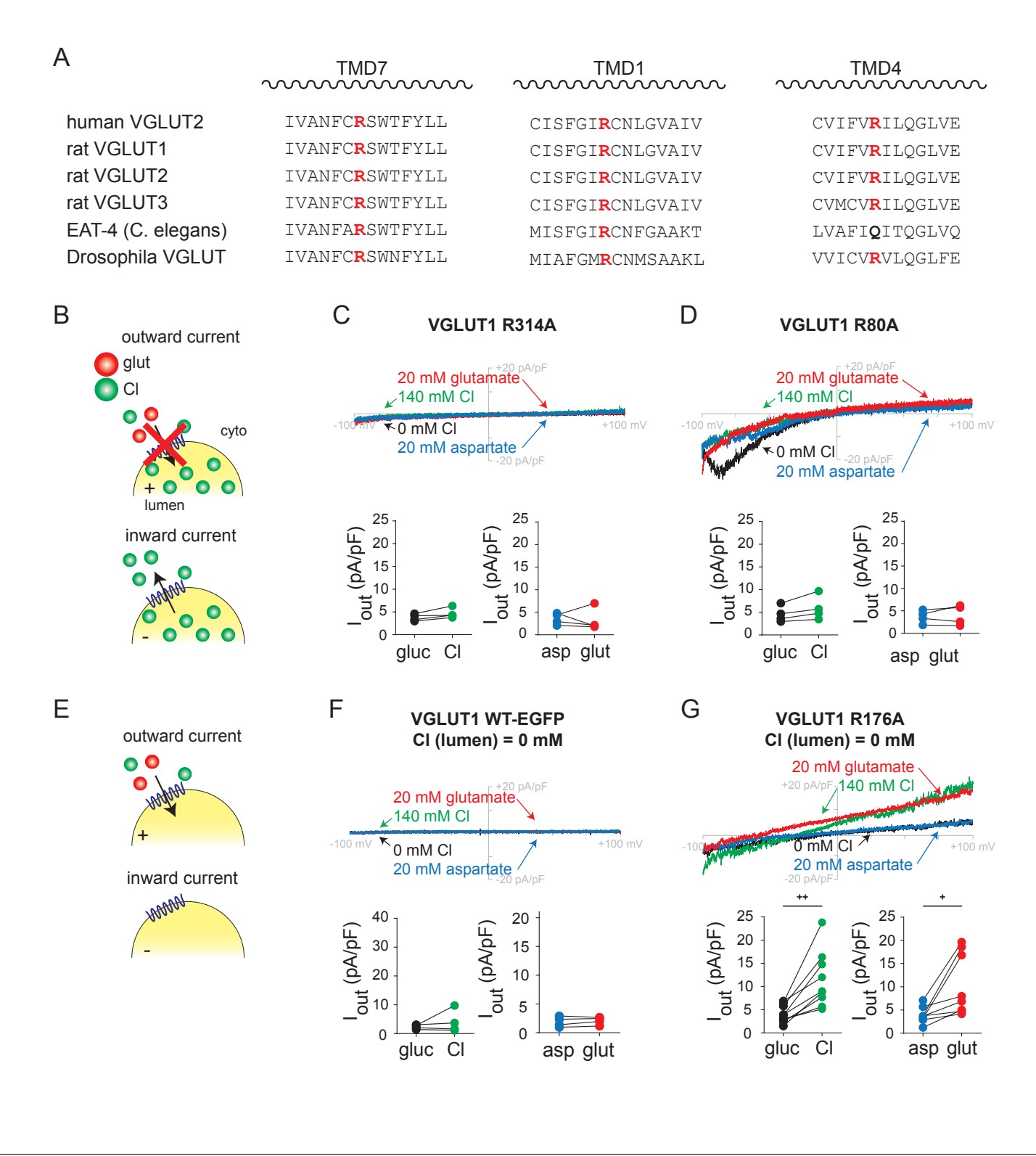

**Figure 3.** Transmembrane arginine residues control allosteric activation and permeation by Cl⁻. (**A**) Sequence alignment of transmembrane domains TM 7, 1 and 4 from the related VGLUT isoforms and species with highly conserved arginine highlighted (red). (**B**) Representative whole endosome recordings (with 140 mM NMDG Cl at pH 5.0 in the pipette) of VGLUT1 R314A (**C**) and R80A (**D**) (n = 4 each). (**E**) Representative recordings with 0 mM Cl⁻ in the pipette of endosomes expressing VGLUT1 WT (**F**) and R176A (**G**) (n = 5–9). Insets show maximum outward currents in the different external

*Figure 3 continued on next page*

Figure 3 continued

solutions (C, $I_{Cl}$, p=0.547, $I_{glut}$, p=0.080 both by paired t-test, D, $I_{Cl}$, p=0.062, $I_{glut}$, p=0.782 both by paired t-test, F, $I_{Cl}$, p=0.625 by Wilcoxon, $I_{glut}$, p=0.397 by paired t-test, G, $I_{Cl}$, p=0.002, $I_{glut}$, p=0.017 both by paired t-test). +p<0.05 and ++p<0.01 by paired t-test. Insets (**C,D,F,G**) indicate maximal outward currents for each endosome.

DOI: https://doi.org/10.7554/eLife.34896.009

The following figure supplements are available for figure 3:

**Figure supplement 1.** Chloride and glutamate currents in endosomes expressing mutant VGLUTs.

DOI: https://doi.org/10.7554/eLife.34896.010

**Figure supplement 2.** Chloride and glutamate currents in endosomes expressing TM4 arginine mutants.

DOI: https://doi.org/10.7554/eLife.34896.011

**Figure supplement 3.** R184A eliminates the requirement for external $Cl^-$ to activate the inward $Cl^-$ conductance of VGLUT2 misexpressed at the plasma membrane.

DOI: https://doi.org/10.7554/eLife.34896.012

## Chloride and glutamate translocate using distinct mechanisms but compete for permeation

Many of the same factors (such as $\Delta\psi$ and lumenal $Cl^-$) activate both glutamate and $Cl^-$ currents, raising questions about the relationship between the two permeant anions. To compare the pathways for permeation, we examined the dose-response to external $Cl^-$ (*Figure 4A*). The outward $Cl^-$ currents show no evidence of saturation up to 140 mM (*Figure 4B*). In contrast, the glutamate currents saturate with $EC_{50}$376 µM (204–726 µM 95% confidence interval) in 10 mM external $Cl^-$ (*Figure 4C*), an apparent affinity slightly higher than previously measured (1–3 mM) using radiotracer flux assays (*Naito and Ueda, 1985*; *Tabb et al., 1992*) but very similar to that determined in the absence of $Cl^-$ or from glutamate-induced acidification (0.2–0.3 mM) (*Disbrow et al., 1982*; *Maycox et al., 1988*). The two anions thus appear to permeate through entirely different mechanisms, $Cl^-$ through the low affinity conduction pathway of a channel, and glutamate through the alternating access mechanism of a transporter, similar to the EAATs (*Fairman et al., 1995*; *Wadiche et al., 1995*).

On the other hand, the common features of $Cl^-$ and glutamate currents, such as the allosteric requirement for lumenal $Cl^-$, suggest a shared mechanism. For this reason, we repeated the dose-response curve to glutamate, but in the presence of 1 mM rather than 10 mM $Cl^-$ since both concentrations have been reported to confer allosteric activation of vesicular glutamate transport (*Naito and Ueda, 1985*). Remarkably, lowering external $Cl^-$ to 1 mM shifts the glutamate dose-response dramatically to the left, with Km 2.95 µM (0.8–10.3 µM 95% confidence interval) (*Figure 4C*), two orders of magnitude higher apparent affinity than previously measured using either radiotracer flux or vesicle acidification. The response to glutamate also differs in cooperativity, with Hill coefficient 0.51 ± 0.14 in 1 mM $Cl^-$ and 1.41 ± 0.37 in 10 mM $Cl^-$, possibly contributing to the dramatic effect of $Cl^-$ on substrate recognition. The effect of $Cl^-$ on glutamate transport demonstrates competition between the two anions for permeation through the VGLUTs. The two anions thus use a related permeation pathway, but differ in the mode of permeation.

## Replacement of lumenal chloride with glutamate fails to confer allosteric activation

To determine how the VGLUTs function late in the process of synaptic vesicle filling, when glutamate replaces lumenal $Cl^-$, we substituted pipette $Cl^-$ with glutamate. In contrast to high lumenal $Cl^-$, which confers both inward and outward flux of $Cl^-$ (*Figure 4E*), high lumenal glutamate does not enable $Cl^-$ influx (*Figure 4F*). The absence of inward as well as outward currents with high lumenal glutamate also shows that glutamate cannot undergo efflux under these conditions. The loss of allosteric activation thus effectively inactivates the VGLUTs, preventing the efflux of glutamate after vesicle filling.

## Discussion

By controlling the ionic gradients independent of $\Delta\psi$, whole endosome recording shows that $Cl^-$ interacts with the VGLUTs as both permeant ion and allosteric activator. As previously predicted from vesicle acidification and oocyte recording (*Bellocchio et al., 2000*; *Schenck et al., 2009*;

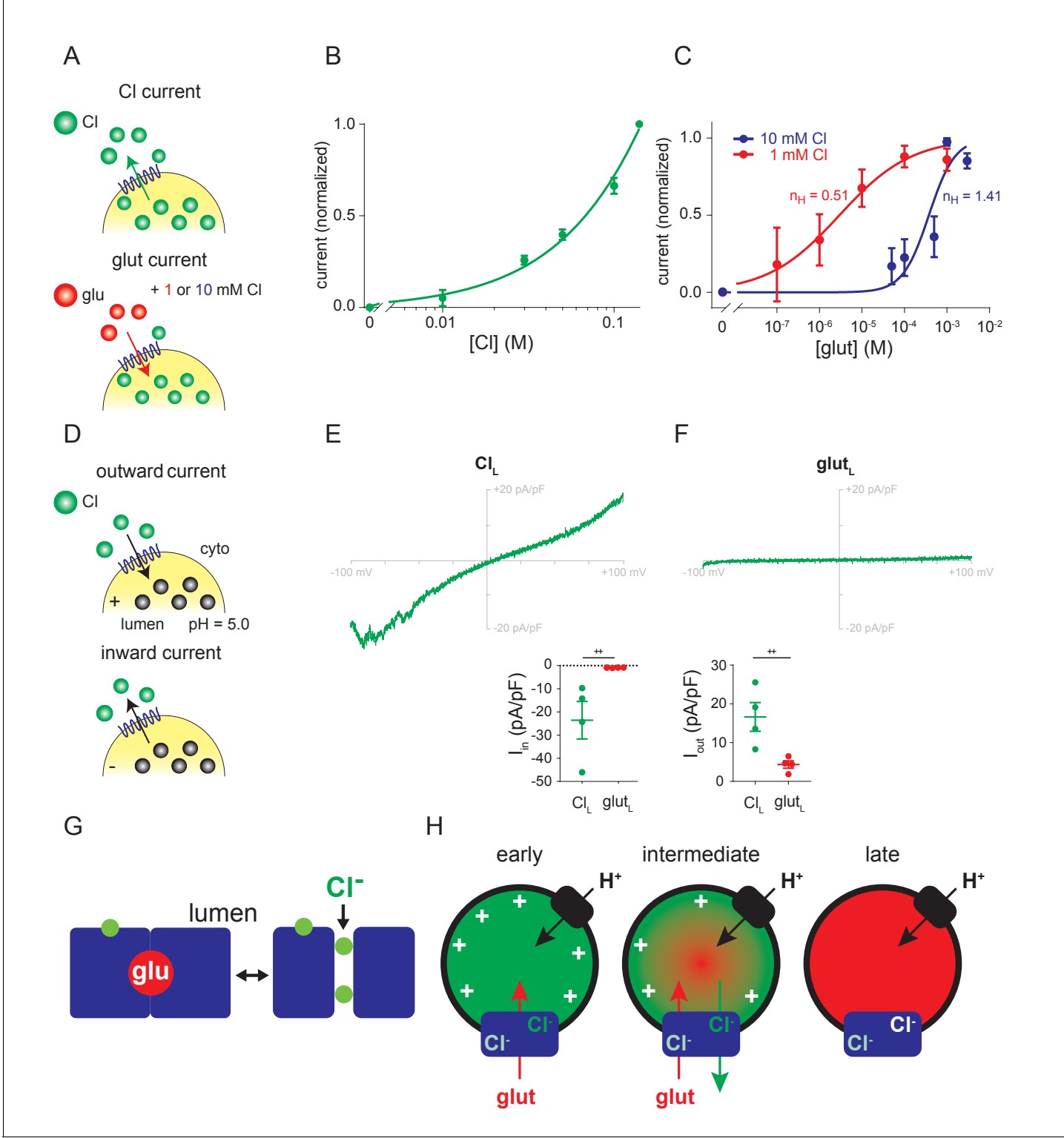

**Figure 4.** Relationship between Cl⁻ and glutamate conductances. (A–C) Dose-response of outward currents to external Cl⁻ (B) or glutamate in the presence of either 1 mM or 10 mM external Cl⁻ (C), all at lumenal pH 5.0. The $EC_{50}$ for glutamate (C) was 376 μM in 10 mM Cl⁻ and 2.95 μM in 1 mM Cl⁻ (p=0.017 by Mann-Whitney, n = 5–6 each). $n_H$, the Hill coefficients for glutamate in different cytoplasmic Cl concentrations. (D–F) Representative recordings and compiled data for endosomes expressing VGLUT1 with either high lumenal Cl⁻ (E) or glutamate (F) in the presence of 140 mM external Cl⁻ ($I_{in}$, p=0.029, $I_{out}$, p=0.029 both by Mann-Whitney) (++p<0.05 by Mann-Whitney test). (G) Permeation of glutamate by an alternating access mechanism, with the occluded state shown on the left, and permeation by Cl⁻ through a channel (shown on the right). Both anions use a related

*Figure 4 continued on next page*

*Figure 4 continued*

permeation pathway, but the arginine in TM4 that confers allosteric activation by lumenal Cl⁻ lies outside the permeation pathway. (**H**) Model for different stages in the filling of synaptic vesicles with glutamate. Immediately after endocytosis (early), the synaptic vesicle contains high concentrations of Cl⁻ which provide the allosteric activation required for anion flux by the VGLUTs (blue). The H⁺ pump (black) provides the driving force Δψ and cytosolic Cl⁻ allosterically activates rather than inhibits due to the saturating concentration of cytosolic glutamate. As glutamate enters (intermediate), Δψ dissipates and the vesicle acidifies, activating the Cl⁻ conductance and the resulting Cl⁻ efflux maintains the Δψ that drives glutamate uptake. When vesicle filling approaches completion (late), the different permeation mechanisms ensure that efflux of Cl⁻ but not glutamate maintains Δψ while stabilizing the accumulated transmitter.

DOI: https://doi.org/10.7554/eLife.34896.013

*Preobraschenski et al., 2014*; *Eriksen et al., 2016*), the VGLUTs exhibit a Cl⁻ conductance, with both inward currents due to the efflux of Cl⁻ and outward currents due to Cl⁻ influx, and a shift of reversal potential in the appropriate direction. The bidirectional flux contrasts with the strong rectification observed at the plasma membrane of *Xenopus* oocytes (*Eriksen et al., 2016*) and supports a physiological role for Cl⁻ efflux as well as entry. Indeed, we previously found that in the absence of a H⁺ pump, an outwardly directed Cl⁻ gradient can drive glutamate uptake by proteoliposomes reconstituted with VGLUT2 (*Eriksen et al., 2016*). The associated Cl⁻ conductance can thus provide the driving force for glutamate uptake, and recent work has indeed suggested that Cl⁻ efflux early after endocytosis can influence activity of the H⁺ pump, presumably by promoting Δψ (*Martineau et al., 2017*).

In addition to the allosteric activation of vesicular glutamate transport by cytosolic Cl⁻ reported using synaptic vesicles (*Naito and Ueda, 1985*; *Maycox et al., 1988*; *Tabb et al., 1992*; *Hartinger and Jahn, 1993*; *Wolosker et al., 1996*; *Juge et al., 2010*), endosome recording now documents the requirement of glutamate transport for lumenal Cl⁻. Originally suggested on the basis of VGLUT coreconstitution with a H⁺ pump that might have reflected an effect of lumenal Cl⁻ on Δψ (*Schenck et al., 2009*; *Martineau et al., 2017*), the use of voltage clamp to measure currents at the plasma membrane demonstrated allosteric regulation of the Cl⁻ conductance (*Eriksen et al., 2016*). Endosome recording now demonstrates that allosteric regulation by lumenal Cl⁻ extends to glutamate transport. Remarkably, the analysis of mutations shows that lumenal activation appears to depend on the interaction of Cl⁻ with a single, highly conserved arginine in TM4. This does not exclude a role for other residues in the recognition of lumenal Cl⁻ but neutralization of the TM4 arginine appears sufficient to confer the activation normally provided by Cl⁻. The results predict that this basic residue normally faces the vesicle lumen and that in the absence of lumenal Cl⁻, it prevents both vesicular glutamate transport and the associated Cl⁻ conductance.

Comparison of the currents due to Cl⁻ and glutamate supports a fundamental difference in the mode of permeation. The Cl⁻ currents do not saturate with increasing concentration of external Cl⁻, consistent with permeation through an ion channel. In contrast, glutamate saturates with a Km ~0.5 mM (in 10 mM Cl⁻), very similar to that observed for glutamate uptake by synaptic vesicles and for synaptic vesicle acidification by glutamate (*Hnasko et al., 2010*). Despite these differences, Cl⁻ permeation and glutamate flux both depend on lumenal Cl⁻ for allosteric activation (as well as on Δψ for the driving force).

Consistent with similar requirements for activation of the conductances, endosome recording now shows that Cl⁻ and glutamate compete for permeation. The inhibition of glutamate uptake by high concentrations of Cl⁻ has generally been interpreted as reflecting dissipation of the driving force Δψ. Recording endosome currents under voltage clamp now shows that Cl⁻ exerts a dramatic effect on glutamate permeation independent of Δψ. The apparent affinity for glutamate shifts by ~100 fold despite only a 10-fold change in Cl⁻ concentration, raising the possibility of multiple Cl⁻ binding sites. Consistent with this possibility, Cl⁻ allosterically activates the VGLUTs from both cytoplasmic and lumenal sides of the membrane. The cooperativity for glutamate also differs in 1 and 10 mM Cl⁻, further supporting the possibility of interaction at two sites. The precise relationship between the Cl⁻ and glutamate remains unclear and they permeate using distinct mechanisms, but the two anions thus appear to use a similar permeation pathway (*Figure 4G*) subject to the same regulation by lumenal Cl⁻, through neutralization of the arginine in TM4. Although Cl⁻ does not influence glutamate transport by the plasma membrane EAATs, they also exhibit an uncoupled Cl⁻ conductance (*Wadiche et al., 1995*; *Arriza et al., 1997*). In contrast to the VGLUTs, however, recent work on the

EAATs has suggested that the permeation pathway for Cl⁻ is entirely distinct from that for glutamate (*Ryan and Mindell, 2007*; *Cater et al., 2014*; *Machtens et al., 2015*).

To determine the role of the Cl⁻ conductance in synaptic vesicle filling with glutamate, a recent study used the dye Rose Bengal as a specific inhibitor of glutamate transport but not the VGLUT-associated Cl⁻ conductance (*Martineau et al., 2017*). However, the concentration of dye used to test the effect on Cl⁻ conductance was 10-fold lower than that used to inhibit glutamate transport. We now find that the established VGLUT inhibitor Evans Blue reduces both Cl⁻ and glutamate currents with a similar IC$_{50}$ ~0.2 μM. The results indicate conservation of the Cl⁻ conductance among all VGLUT isoforms, consistent with its importance. However, understanding its physiological role requires more specific tools to inhibit Cl⁻ permeation by the VGLUTs.

What is the relationship between allosteric activation and permeation by Cl⁻? Since neutralization of the lumenally oriented arginine in TM4 confers both Cl⁻ and glutamate currents, this residue cannot be required for permeation (*Figure 4G*). However, we cannot exclude the possibility that cytosolic Cl⁻ allosterically activates from within the permeation pathway.

The manipulation of ionic gradients independent of Δψ at both the plasma membrane and the endosome has also enabled us to isolate different stages in the process of synaptic vesicle filling (*Figure 4H*), in a way that is not possible in the presence of an active H⁺ pump and changes in the ionic gradients across the synaptic vesicle membrane, such as those recently reported for Cl⁻ (*Martineau et al., 2017*). Initially, the Cl⁻ trapped in recycling vesicles by endocytosis confers the allosteric activation required for glutamate transport. In addition to the H⁺ pump, the efflux of lumenal Cl⁻ might contribute to the Δψ that drives glutamate uptake, but since the Cl⁻ conductance associated with VGLUTs requires allosteric activation by low lumenal pH, other anion carriers may be responsible for the Cl⁻ efflux that immediately follows endocytosis (*Eriksen et al., 2016*; *Martineau et al., 2017*). At the same time, cytosolic Cl⁻ (~20 mM [*Price and Trussell, 2006*]) might be expected to compete with glutamate for permeation. However, cytosolic Cl⁻ also provides allosteric activation of the VGLUTs and the outwardly directed Cl⁻ gradient will impede Cl⁻ influx. In addition, the higher affinity of VGLUTs for glutamate predicts that ~10 mM cytosolic glutamate (*Storm-Mathisen and Ottersen, 1990*) should produce maximal uptake despite even higher cytosolic Cl⁻. As glutamate entry dissipates Δψ, disinhibiting the H⁺ pump, the resulting drop in lumenal pH activates the Cl⁻ conductance associated with VGLUTs (*Eriksen et al., 2016*). The robust inward currents observed by endosome recording suggest that the resulting Cl⁻ efflux will support Δψ despite the accumulation of lumenal H⁺, and so maintain the driving force for glutamate uptake. The differences in permeation by Cl⁻ and glutamate would further serve to retain the glutamate as vesicles fill, but allow Cl⁻ efflux to sustain Δψ. Indeed, we previously found that synaptic vesicles acidified in glutamate exhibit a more stable ΔpH than vesicles acidified in Cl⁻ (*Hnasko et al., 2010*) and these results indicate the mechanism: the loss of lumenal H⁺ requires loss of lumenal anion for charge balance, and glutamate uses an alternating access mechanism whereas Cl⁻ permeates through channel. Finally, the replacement of lumenal Cl⁻ by glutamate effectively inactivates the VGLUTs, preventing the leakage of glutamate from filled vesicles.

Taken together, the results suggest that the Cl⁻ conductance associated with vesicular glutamate transport serves to maintain Δψ despite the accumulation of ΔpH, and allosteric regulation serves to coordinate flux of the two anions.

## Materials and methods

### Key resources table

| Reagent type | Designation | Source or reference | Identifiers | Additional |
|---|---|---|---|---|
| Gene (*Rattus norvegicus*) | VGLUT1 | Edwards lab | NM_053859.2 | |
| Gene (*Rattus norvegicus*) | VGLUT2 | Edwards lab | NM_053427.1 | |
| Gene (*Rattus norvegicus*) | VGLUT3 | Edwards lab | NM_153725.1 | |
| Cell line (*Homo sapiens*) | HEK293T cells | UCSF Cell Culture Fac. | | |
| Transfected construct | pEGFP-N1 | Clontech | | |
| Antibody | Mouse anti-actin | Sigma | MAB1501R | (1:3000) |

*Continued on next page*

*Continued*

| Reagent type | Designation | Source or reference | Identifiers | Additional |
|---|---|---|---|---|
| Antibody | Guinea pig anti-VGLUT1 | Millipore Sigma | AB5905 | (1:2000) |
| Recombinant DNA reagent | PolyJet | SignaGen | SL100688 | |
| Recombinant DNA reagent | Lipofectamine 2000 | Gibco Life Technologies | | |
| Software, algorithm | Prism 5.0 | GraphPad | | |

## Constructs

All three isoforms of the *Rattus norvegicus* vesicular glutamate transporters (NCBI Reference Sequences: NM_053859.2, NM_053427.1, NM_153725.1) were amplified by PCR using primers designed to include a Kozak sequence (5'-CGCCACC-3') at the 5' end as well as the necessary restriction sites for subcloning: NheI/AflII for pVGLUT1- and pVGLUT2-EGFP and NheI/AgeI for pVGLUT3-EGFP. The products were then subcloned into the pEGFP-N1 vector (Clontech). Alanine was substituted for the arginines in TM7, 1 and 4 of rat VGLUT cDNAs using either QuickChange or New England Biolabs' Q5 Site-Directed Mutagenesis Kit (Catalog No. E0554), and the change confirmed by DNA sequencing. The Rab5 Q79L-mCherry construct (AddGene Plasmid No. 35138), was used to enlarge endosomes.

## Cell culture

HEK293T cells were cultured in Dulbecco's Modified Eagle Media (ThermoFisher, Catalog No. 11965–092, Waltham, MA.) with 10% defined Fetal Bovine Serum (GE Healthcare Life Sciences HyClone, Catalog No. SH30070.0, Pittsburgh, PA.) and were independently authenticated and confirmed negative for mycoplasma (both by the ATCC). Rab5 Q79L-mCherry and wild type or mutant pVGLUT-EGFP (1 μg each) were cotransfected into HEK293T cells (~70–90% confluence) in 6-well tissue culture plates using the PolyJet transfection reagent (SignaGen Catalog No. SL100688, Rockville, MD.) at a ratio of 3:1 (PolyJet (μL):DNA (μg)). Following the manufacturer's recommendation, we incubated the PolyJet transfection complex for 12–18 hr, then replaced the solution with fresh serum-containing media. After 4–5 hr, the transfected cells were trypsinized and replated onto poly-L-lysine-coated coverslips in the same serum-containing media, and incubated until recording the next day.

## Electrophysiology

### Whole endosome recording

Borosilicate glass patch pipettes (Sutter Instruments, Catalog No. BF150-75-10) were made with a Sutter Instrument P-87 puller (Novato, CA.) and fire-polished with the Narishige Microforge MF-830 (Amityville, NY.) to a final resistance of 6–10 MΩ. As previously described (*Cang et al., 2015*), the plasma membrane of HEK293T cells containing enlarged endosomes was ruptured with a glass pipette. The enlarged endosome was then extruded using the same pipette tip. Another clean pipette tip was then used to patch on and form a gigaohm seal with the isolated endosome. Only one enlarged endosome was recorded from each coverslip. Patch recordings were performed with an Axopatch 200B amplifier (Molecular Devices, Sunnyvale, CA.) and a DigiData 1440A data acquisition system (Molecular Device, Sunnyvale, CA.). The recordings obtained from the amplifier were not inverted. The standard voltage ramp protocol from $-100$ mV to $+100$ mV lasted 800 msecs. Before beginning the ramp, the holding potential at 0 mV was dropped to $-100$ mV for 50 msec. The standard bath (cytosolic) solution contained (in mM) 140 NMDG gluconate, 2 $MgSO_4$, 20 HEPES, 5 EGTA, 10 glucose, pH 7.2. The standard pipette (luminal) solution contained (in mM) 140 NMDG Cl, 1 $MgCl_2$, 20 MES, 10 Glucose, 2 $CaCl_2$, pH to 5.0. Substitutions to these standard solutions are noted in the figure legends. To manipulate pH, we used either 20 mM HEPES, pH 7.2 or 20 mM MES, pH 5.0. Addition of L-glutamic acid or L-aspartic acid involved replacement of equimolar gluconate to maintain osmolarity. Similarly, $Cl^-$, $Br^-$ and $PO_4$ replaced gluconate. Figures were plotted with MATLAB (Mathworks, Natick, MA.), Adobe Illustrator (Adobe, San Jose, CA.), and Prism (GraphPad, LaJolla, CA.). Nonlinear regression was used to fit the dose-response curves.

## Whole cell recording

One to two days before recording, HEK293T cells were transfected using Lipofectamine 2000 (Life Technologies) with 2 μg pIRES2-EGFP or pIRES-EGFP FI/AA 2 × GG VGLUT2 per 700,000 cells. The cells were trypsinized 1–4 hr before recording, and plated onto polylysine-coated glass coverslips. Whole-cell patch clamp recordings of EGFP-positive cells were performed using an Axopatch 200B amplifier (Molecular Devices) at room temperature. Electrodes pulled from borosilicate glass (BF150-86-7.5; Sutter Instruments) were fire polished and back filled with pipette solution containing 145 mM TMA-gluconate, 30 mM HEPES pH 7.5, 5 mM EGTA, 2.5 mM $MgCl_2$, ±5 mM glutamate (mOsm 315–320). Gigaohm seals were formed in Ringer's bath solution containing (in mM) 130 NaCl, 10 HEPES pH 7.4, 10 glucose, 4 KCl, 2 $CaCl_2$, 1 $MgCl_2$. A 3 M KCl agar salt bridge was used for bath reference electrode. Access resistance and membrane capacitance were determined using the Membrane Test application of pClamp 10 (Molecular Devices). Currents were recorded using a 800 ms ramp from −100 mV to 60 mV in the following bath solutions (all 320–325 mOsm): Chloride, pH 7.5 (in mM, 145 choline $Cl^-$, 30 HEPES, 2 Mg gluconate), Chloride, pH 5.5 (145 choline $Cl^-$, 50 MES, 2 Mg gluconate), Gluconate, pH 7.5 (145 choline gluconate, 30 HEPES, 2 Mg gluconate) and Gluconate solution, pH 5.5 (145 choline gluconate, 50 MES, 2 Mg gluconate). The currents were sampled at 10 kHz and low pass filtered at 1 kHz.

## Quantitative western analysis

Two days after transfection, the HEK293T cells were lysed for 30 min at 4°C in buffer containing 25 mM Tris, pH 7.5, 150 mM NaCl, 1% Triton X-100, 1 mM EDTA supplemented with complete protease inhibitor (Roche), the extract sedimented for 20 mins at 14000 g at 4°C and the pellet discarded. Five mg supernatant protein was separated by electrophoresis through 10% SDS-polyacrylamide and transferred to a nitrocellulose membrane. The membrane was blocked in 150 mM NaCl with 50 mM Tris, pH 7.4 (TBS) and 5% dry milk, incubated with guinea pig anti-VGLUT1 (1:2000, Millipore Sigma) or mouse anti-actin (1:3000, Sigma) in TBS + 0.1% Tween-20 overnight at 4°C, washed three times for 5 min each in TBS + 0.1% Tween-20, incubated for 30 min at room temperature with IRDye 800CW Donkey anti-Guinea Pig or IRDye 800CW Goat anti-Mouse IgG (1:20000, LI-COR) in blocking buffer + 0.1% Tween-20, washed three times for 10 min each in TBS + 0.1% Tween-20 and rinsed in TBS. The membrane was scanned using an Odyssey device (LI-COR) and the resulting images quantified in ImageJ.

## Statistical analysis

The parametric paired t-test or non-parametric Wilcoxon test were used to determine statistical significance between currents obtained from the same endosome in different external solutions, depending on whether the distribution of individual data was Gaussian. The Mann-Whitney test was used to compare the EC50 for glutamate of endosomes with different external $Cl^-$ concentrations.

## Acknowledgements

We thank C Cang and D Ren (UPenn) and Y Kirichok (UCSF) for assistance with the whole endosome recording, and R Swanson, L Jan, T Logan, and members of the Edwards lab for helpful discussions. This work was supported by NIH fellowship F31 NS092298-01A1 and training grant T32 GM008568 (to RC), postdoctoral fellowship R77-6780 from the Lundbeck Foundation (to JE) and Sandler Fund in Basic Sciences and NIH grants R37 MH50712 and 5R01 NS089713 (to RHE).

## Additional information

### Funding

| Funder | Grant reference number | Author |
|---|---|---|
| National Institute of Neurological Disorders and Stroke | T32 GM008568 | Roger Chang |
| National Institutes of Health | F31 NS09229-01A1 | Roger Chang |
| Lundbeckfonden | R77-6780 | Jacob Eriksen |

| National Institute of Mental Health | R37 MH50712 | Robert H Edwards |
|---|---|---|
| Sandler Foundation | Basic Sciences | Robert H Edwards |
| National Institutes of Health | 5R01 NS089713 | Robert H Edwards |
| National Institutes of Health | 5R01 NS08971 | Robert H Edwards |

The funders had no role in study design, data collection and interpretation, or the decision to submit the work for publication.

### Author contributions

Roger Chang, Data curation, Formal analysis, Funding acquisition, Validation, Investigation, Visualization, Methodology, Writing—original draft, Writing—review and editing; Jacob Eriksen, Investigation, Data curation, Formal analysis, Writing—review and editing; Robert H Edwards, Conceptualization, Resources, Data curation, Formal analysis, Supervision, Funding acquisition, Investigation, Methodology, Writing—original draft, Writing—review and editing

### Author ORCIDs

Roger Chang https://orcid.org/0000-0001-5654-2480
Jacob Eriksen https://orcid.org/0000-0003-0726-9769
Robert H Edwards http://orcid.org/0000-0002-8650-3260

### Decision letter and Author response

Decision letter https://doi.org/10.7554/eLife.34896.016
Author response https://doi.org/10.7554/eLife.34896.017

## Additional files

### Supplementary files

• Transparent reporting form
DOI: https://doi.org/10.7554/eLife.34896.014

### Data availability

All data generated or analysed during this study are included in the manuscript and supporting files.

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
