## [Decision Letter]

Thank you for submitting your article "The Dual Role of Chloride in Vesicular Glutamate Transport" for consideration by *eLife*. Your article has been favorably evaluated by Richard Aldrich (Senior Editor) and three reviewers, one of whom is a member of our Board of Reviewing Editors. The following individual involved in review of your submission has agreed to reveal their identity: Christof Grewer (Reviewer #2).

The reviewers have discussed the reviews with one another and the Reviewing Editor has drafted this decision to help you prepare a revised submission.

This manuscript from the Edwards lab presents a very nice, and important new approach to studying the VGLUT vesicular glutamate transporters. The authors take advantage of a method, developed for studying organellar channels, to patch clamp genetically enlarged endosomes that express the VGLUT transporter. The data shown here demonstrate the method and provide convincing, well-controlled arguments that the measured currents indeed reflect the activity of VGLUT transporters. Using the method, the authors then tackle a long-standing vexation in the field, the effects of Cl^-^ on the transport process. Previous work suggests that the role of Cl^-^ is complex, but the previously available experimental systems have not had the resolution to definitively separate the multiple effects of Cl. Here, though, the authors present strong evidence for both an uncoupled Cl^-^ conductance and an allosteric activation of glutamate transport by Cl^-^. Furthermore, they show that Cl^-^ competes with glutamate for the transport pathway, for the first time convincingly arguing that the pathways for the two substrates overlap. Finally, mutations of several arginine residues have differing effects on VGLUT properties, with one of them (R176) seeming to abolish the requirement for luminal Cl^-^ to activate outward currents.

All three reviewers expressed enthusiasm for the work. There were, however, several concerns that the authors need to address.

Essential revisions:

1) It should be possible to convert the currents in response to voltage ramps to actual I-V plots. This would make it much easier for the reader to assess reversal potentials and whether the current are rectifying or not.

2) Interpretation of data in Figure 4C is problematic. If Cl^-^ and glutamate competed for the same binding site, then 10 fold change in Cl^-^ concentration should lead to 10-fold change in the affinity for Glu, not 100. Either there are two Cl ions competing with glutamate cooperatively or something else is going on. Additionally, what causes the much higher cooperativity of the glutamate dose response at 10 mM chloride vs. 1 mM chloride? The Hill coefficients should be stated in the text.

3) Is there a reason why Evans Blue does not result in complete inhibition of the VGLUT currents?

4) How is the expression level of VGLUTs controlled? Is the quantification in Figure 1—figure supplement 1C and Figure 1—figure supplement 2D, where the currents through distinct VGLUTs are compared valid? Similarly, how can currents through mutant transporters be compared if they are not controlled quantitatively for protein expression?

5) Luminal Cl^-^ produces allosteric activation with very low apparent affinity in 50 mM range. That seems quite low for a binding reaction, though possible. It would be important to show that this is, indeed a saturable process and to construct a complete activation curve.

6) Potentiation of glutamate currents in the presence of 10 mM external chloride is less than 2 fold. This hardly can be called "great" potentiation. In fact, it almost appears that the glutamate and Cl^-^ currents are additive. Could that be the case? How would one establish glutamate that the cytoplasmic Cl^-^ potentiates glutamate transport through an allosteric effect?

7) I do not see the data showing that Br^-^ activates glutamate currents. Figure 2—figure supplement 1 does not seem to show this. In Figure 2—figure supplement 1, the reversal potential changes going from 20 mM glutamate to 140 mM Br^-^ are vastly different for distinct VGLUTs, which is confusing.

8) Mutation in TM4 appears to effectively inhibit inward currents. Thus, it must affect the Cl^-^ permeation pathway. In fact, it is difficult to understand how the same mutation inhibits Cl^-^ current in one direction (outward) but not the other (inward).

9) Generally speaking, the data on TM4 mutants seem limited. The conclusion that neutralization of R176 is the mechanism of Cl^-^ allosteric action seems premature. At the very least, one would consider making more subtle mutations, such as R to Q and R to K to further validate the hypothesis.

---

## [Author Response]

Essential revisions:1) It should be possible to convert the currents in response to voltage ramps to actual I-V plots. This would make it much easier for the reader to assess reversal potentials and whether the current are rectifying or not.

As mentioned before, the variability in inward currents makes precise determination of reversal potential unreliable, but we have now added y- as well as x-axes to provide more information about the I-V relationship. Rather than average the ramps from multiple endosomes, we still prefer to present a representative endosome and show the scatterplot of currents from different endosomes.

2) Interpretation of data in Figure 4C is problematic. If Cl^-^ and glutamate competed for the same binding site, then 10 fold change in Cl^-^ concentration should lead to 10-fold change in the affinity for Glu, not 100. Either there are two Cl ions competing with glutamate cooperatively or something else is going on. Additionally, what causes the much higher cooperativity of the glutamate dose response at 10 mM chloride vs. 1 mM chloride? The Hill coefficients should be stated in the text.

We were also surprised by the large difference in glutamate affinity at 1 and 10 mM Cl^-^. Cooperativity for Cl^-^ is distinctly possible, and there appear to be two sites for allosteric activation by Cl^-^, one cytosolic and the other lumenally oriented. We have now included this point in the text of the Results and Discussion. We have also included Hill coefficients for glutamate under the two conditions—they do show a substantial difference in cooperativity which might be related to the large shift in affinity due to Cl^-^.

3) Is there a reason why Evans Blue does not result in complete inhibition of the VGLUT currents?

We presume that this reflects endogenous (background) conductances that would not be inhibited by Evans Blue.

4) How is the expression level of VGLUTs controlled? Is the quantification in Figure 1—figure supplement 1C and Figure 1—figure supplement 2D, where the currents through distinct VGLUTs are compared valid? Similarly, how can currents through mutant transporters be compared if they are not controlled quantitatively for protein expression?

All of the recordings were made from GFP^+^ endosomes, suggesting that we recorded from endosomes that all express significant amounts of VGLUT protein. We agree that fluorescence does not provide a quantitative estimate of expression, but the isoforms do not show major differences in the currents (Figure 1—figure supplements 1 and 2, Figure 2—figure supplement 2), reducing the likelihood of a major difference in expression. In the case of point mutants, we have nonetheless used quantitative western analysis to assess average expression for each mutant construct. Figure 3—figure supplement 2F now shows both a representative immunoblot (above) and quantitation of multiple transfections (below). The results indicate expression equivalent to wild type of the inactive R314A (TM7) mutant, and expression by the active R176K mutant similar to the inactive R80A and R176Q. In addition to the use of GFP^+^ endosomes, average expression of the different constructs thus does not correlate with activity.

5) Luminal Cl^-^ produces allosteric activation with very low apparent affinity in 50 mM range. That seems quite low for a binding reaction, though possible. It would be important to show that this is, indeed a saturable process and to construct a complete activation curve.

This apparent affinity resembles activation of the Cl^-^ conductance in *Xenopus* oocytes by external Cl^-^: low concentrations confer activation, but activation increases substantially at mid-millimolar Cl^-^ (Eriksen et al., 2016). Although comparison of different lumenal contents requires different endosomes, we have now added Figure 2—figure supplement 1 which presents the inward and outward currents in different external Cl^-^. The inward currents do not saturate, presumably because they represent efflux of Cl^-^ and higher lumenal Cl^-^ confers larger efflux and inward currents. However, the outward currents do not increase from 50 to 140 mM, consistent with a role for lumenal Cl^-^ in allosteric activation rather than permeation.

6) Potentiation of glutamate currents in the presence of 10 mM external chloride is less than 2 fold. This hardly can be called "great" potentiation. In fact, it almost appears that the glutamate and Cl^-^ currents are additive. Could that be the case? How would one establish that the cytoplasmic Cl^-^ potentiates glutamate transport through an allosteric effect?

We agree that the measurement of currents makes it difficult to distinguish which of the two ions is responsible. This is why we included Figure 2G. This shows that 10 mM Cl^-^ has a much bigger effect on outward current magnitude in the presence of glutamate than its absence, excluding additivity and arguing for allostery.

7) I do not see the data showing that Br^-^ activates glutamate currents. Figure 2—figure supplement 1 does not seem to show this. In Figure 2—figure supplement 1, the reversal potential changes going from 20 mM glutamate to 140 mM Br^-^ are vastly different for distinct VGLUTs, which is confusing.

We have interpreted the results with lumenal Br^-^ in light of the finding that 0 and 10 mM lumenal Cl^-^ do not suffice to confer the outward Cl^-^ and glutamate currents (Figure 2 and Figure 2—figure supplement 1). Both glutamate and Br^-^ shift reversal potential in the negative direction, but the extent varies depending on the magnitude of inward currents, making it difficult to draw conclusions from the extent of shift, as noted in point #1 above.

8) Mutation in TM4 appears to effectively inhibit inward currents. Thus, it must affect the Cl^-^ permeation pathway. In fact, it is difficult to understand how the same mutation inhibits Cl^-^ current in one direction (outward) but not the other (inward).

Figure 3G (R176A) shows minimal inward currents because the lumen contains no permeant anion. External Cl^-^ still produces outward currents, so the permeation pathway appears intact. Perhaps the concern is instead with Figure 3D, which shows persistent inward but no outward currents with the R80A mutant. It is difficult to understand how the same mutation might affect flux unidirectionally, but it is also more problematic to evaluate the constitutive inward currents than the outward currents due to addition of Cl^-^ or glutamate. The effect of this mutation thus remains less certain than the others, but the effect nonetheless persists in all three isoforms, and we have included these considerations in the text of the Results.

9) Generally speaking, the data on TM4 mutants seem limited. The conclusion that neutralization of R176 is the mechanism of Cl^-^ allosteric action seems premature. At the very least, one would consider making more subtle mutations, such as R to Q and R to K to further validate the hypothesis.

As suggested, we have now analyzed both the charge-preserving R176K and neutralizing R176Q mutations. Figure 3—figure supplement 2B, E now shows that R176K still requires lumenal Cl^-^, consistent with a role for this cationic residue in Cl^-^ recognition. However, R176Q does not confer any currents even with high lumenal Cl^-^ (Figure 3—figure supplement 2C, and the lack of activity makes it impossible to assess Cl^-^ dependence. These two mutants also express at similar levels, making it unlikely that a difference in expression accounts for the difference in activity.